# Computational Doob's $h$-transforms for Online Filtering of Discretely Observed Diffusions

## Abstract

This paper is concerned with online filtering of discretely observed nonlinear diffusion processes. Our approach is based on the fully adapted auxiliary particle filter, which involves Doob's $h$-transforms that are typically intractable. We propose a computational framework to approximate these $h$-transforms by solving the underlying backward Kolmogorov equations using nonlinear Feynman-Kac formulas and neural networks. The methodology allows one to train a locally optimal particle filter prior to the data-assimilation procedure. Numerical experiments illustrate that the proposed approach can be orders of magnitude more efficient than state-of-the-art particle filters in the regime of highly informative observations, when the observations are extreme under the model, and if the state dimension is large.

## 1 Introduction

Diffusion processes are fundamental tools in applied mathematics, statistics, and machine learning. Because this flexible class of models is easily amenable to computations and simulations, diffusion processes are very common in biological sciences (e.g. population and multi-species models, stochastic delay population systems), neuroscience (e.g. models for synaptic input, stochastic Hodgkin–Huxley model, stochastic Fitzhugh–Nagumo model), and finance (e.g. modeling multi assets prices) (Allen, 2010; Shreve et al., 2004; Capasso & Capasso, 2021). In these disciplines, tracking signal from partial or noisy observations is a very common task. However, working with diffusion processes can be challenging as their transition densities are only tractable in rare and simple situations such as (geometric) Brownian motions or Ornstein–Uhlenbeck (OU) processes. This difficulty has hindered the use of standard methodologies for inference and data-assimilation of models driven by diffusion processes and various approaches have been developed to circumvent or mitigate some of these issues, as discussed in Section 4.

Consider a time-homogeneous multivariate diffusion process $d\mathbf{X}_t = \mu(\mathbf{X}_t)\,dt + \sigma(\mathbf{X}_t)\,d\mathbf{B}_t$ that is discretely observed at regular intervals. Noisy observations $\mathbf{y}_k$ of the latent process $\mathbf{X}_{t_k}$ are collected at equispaced times $t_k \equiv k\,T$ for $k \geq 1$. We consider the online filtering problem which consists in estimating the conditional laws $\pi_k(d\mathbf{x}) = \mathbb{P}(\mathbf{X}_{t_k} \in d\mathbf{x}|\mathbf{y}_1, \ldots, \mathbf{y}_k)$, i.e. the filtering distributions, as observations are collected. We focus on the use of Particle Filters (PFs) that approximate the filtering distributions with a system of weighted particles. Although many previous works have relied on the Bootstrap Particle Filter (BPF), which simulates particles from the diffusion process, it can perform poorly in challenging scenarios as it fails to take the incoming observation $\mathbf{y}_k$ into account. The goal of this article is to show that the (locally) optimal approach given by the Fully Adapted Auxiliary Particle Filter (FA-APF) (Pitt & Shephard, 1999) can be implemented. This necessitates simulating a conditioned diffusion process, which can be formulated as a control problem involving an intractable Doob's $h$-transform (Rogers & Williams, 2000; Chung & Walsh, 2006). We propose the *Computational Doob's $h$-Transform* (CDT) framework for efficiently approximating these quantities. The method relies on nonlinear Feynman-Kac formulas for solving backward Kolmogorov equations simultaneously for all possible observations. Importantly, this preprocessing step only needs to be performed once before starting the online filtering procedure. Numerical experiments illustrate that the proposed approach can be orders of magnitude more efficient than the BPF in the regime of highly informative observations, when the observations are extreme under the model, and

if the state dimension is large. A PyTorch implementation to reproduce our numerical experiments is available at `https://anonymous.4open.science/r/CompDoobTransform/`.

**Notations.** For two matrices $A, B \in \mathbb{R}^{d,d}$, their Frobenius inner product is defined as $\langle A, B \rangle_{\mathrm{F}} = \sum_{i,j=1}^{d} A_{i,j} B_{i,j}$. The Euclidean inner product for $\mathbf{u}, \mathbf{v} \in \mathbb{R}^d$ is denoted as $\langle \mathbf{u}, \mathbf{v} \rangle = \sum_{i=1}^{d} u_i v_i$. For two (or more) functions $F$ and $G$, we sometimes use the shortened notation $[FG](\mathbf{x})$ to denote the product $F(\mathbf{x})G(\mathbf{x})$.

## 2 BACKGROUND

### 2.1 FILTERING OF DISCRETELY OBSERVED DIFFUSIONS

Consider a homogeneous diffusion process $\{\mathbf{X}_t\}_{t \geq 0}$ in $\mathcal{X} = \mathbb{R}^d$ with initial distribution $\rho_0(d\mathbf{x})$ and dynamics

$$d\mathbf{X}_t = \mu(\mathbf{X}_t)\,dt + \sigma(\mathbf{X}_t)\,d\mathbf{B}_t, \tag{1}$$

described by the drift and volatility functions $\mu : \mathbb{R}^d \to \mathbb{R}^d$ and $\sigma : \mathbb{R}^d \to \mathbb{R}^{d,d}$. The associated semi-group of transition probabilities $p_s(d\widehat{\mathbf{x}} \mid \mathbf{x})$ satisfies $\mathbb{P}(\mathbf{X}_{t+s} \in A \mid \mathbf{X}_t = \mathbf{x}) = \int_A p_s(d\widehat{\mathbf{x}} \mid \mathbf{x})$ for any $s, t > 0$ and measurable $A \subset \mathcal{X}$. The process $\{\mathbf{B}_t\}_{t \geq 0}$ is a standard $\mathbb{R}^d$-valued Brownian motion. The diffusion process $\{\mathbf{X}_t\}_{t \geq 0}$ is discretely observed at time $t_k = kT$, for $k \geq 1$, for some inter-observation time $T > 0$. The $\mathcal{Y}$-valued observation $\mathbf{Y}_k \in \mathcal{Y}$ at time $t_k$ is modelled by the likelihood function $g : \mathcal{X} \times \mathcal{Y} \to \mathbb{R}_+$ in the sense that for any measurable $A \subset \mathcal{Y}$, we have $\mathbb{P}(\mathbf{Y}_k \in A \mid \mathbf{X}_{t_k} = \mathbf{x}_k) = \int_A g(\mathbf{x}_k, \mathbf{y})\,d\mathbf{y}$ for some dominating measure $d\mathbf{y}$ on $\mathcal{Y}$. For a test function $\varphi : \mathcal{X} \to \mathbb{R}$, the generator of the diffusion process $\{\mathbf{X}_t\}_{t \geq 0}$ is given by $\mathcal{L}\varphi = \langle \mu, \nabla\varphi \rangle + \frac{1}{2} \langle \sigma\sigma^{\top}, \nabla^2\varphi \rangle_{\mathrm{F}}$. This article is concerned with approximating the filtering distributions $\pi_k(d\mathbf{x}) = \mathbb{P}(\mathbf{X}_{t_k} \in d\mathbf{x} \mid \mathbf{y}_1, \ldots, \mathbf{y}_k)$. For notational convenience, we set $\pi_0(d\mathbf{x}) \equiv \rho_0(d\mathbf{x})$ since there is no observation collected at the initial time $t = 0$.

### 2.2 PARTICLE FILTERING

Particle Filters (PF), also known as Sequential Monte Carlo methods, are a set of Monte Carlo algorithms that can be used to solve filtering problems (see Chopin et al. (2020) for a recent textbook on the topic). PFs evolve a set of $M \geq 1$ particles $\mathbf{x}_t^{1:M} = (\mathbf{x}_t^1, \ldots, \mathbf{x}_t^M) \in \mathcal{X}^M$ forward in time using a combination of *propagation* and *resampling* operations. To initialize the PF, each initial particle $\mathbf{x}_0^j \in \mathcal{X}$ for $1 \leq j \leq M$ is sampled independently from the distribution $\rho_0(d\mathbf{x})$ so that $\pi_0(d\mathbf{x}) \approx M^{-1} \sum_{j=1}^{M} \delta(d\mathbf{x}; \mathbf{x}_0^j)$. Approximations of the filtering distribution $\pi_k$ for $k \geq 1$ are built recursively as follows. Given the Monte Carlo approximation of the filtering distribution at time $t_k$, $\pi_k(d\mathbf{x}) \approx M^{-1} \sum_{j=1}^{M} \delta(d\mathbf{x}; \mathbf{x}_{t_k}^j)$, the particles $\mathbf{x}_{t_k}^{1:M}$ are propagated independently forward in time by $\widehat{\mathbf{x}}_{t_{k+1}}^j \sim q_{k+1}(d\widehat{\mathbf{x}} \mid \mathbf{x}_{t_k}^j)$, using a Markov kernel $q_{k+1}(d\widehat{\mathbf{x}} \mid \mathbf{x})$ specified by the user. The BPF corresponds to the Markov kernel $q_{k+1}^{\mathrm{BPF}}(d\widehat{\mathbf{x}} \mid \mathbf{x}) = \mathbb{P}(\mathbf{X}_{t_{k+1}} \in d\widehat{\mathbf{x}} \mid \mathbf{X}_{t_k} = \mathbf{x})$, while the FA-APF (Pitt & Shephard, 1999) corresponds to the (typically intractable) kernel $q_{k+1}^{\mathrm{FA\text{-}APF}}(d\widehat{\mathbf{x}} \mid \mathbf{x}) = \mathbb{P}(\mathbf{X}_{t_{k+1}} \in d\widehat{\mathbf{x}} \mid \mathbf{X}_{t_k} = \mathbf{x}, \mathbf{Y}_{k+1} = \mathbf{y}_{k+1})$. Each particle $\widehat{\mathbf{x}}_{t_{k+1}}^j$ is associated with a normalized weight $\overline{W}_{k+1}^j = W_{k+1}^j / \sum_{i=1}^{M} W_{k+1}^i$, where the unnormalized weights $W_{k+1}^j$ (by time-homogeneity of (1)) are defined as

$$W_{k+1}^j = \frac{p_T(d\widehat{\mathbf{x}}_{t_{k+1}}^j \mid \mathbf{x}_{t_k}^j)}{q_{k+1}(d\widehat{\mathbf{x}}_{t_{k+1}}^j \mid \mathbf{x}_{t_k}^j)} g(\widehat{\mathbf{x}}_{t_{k+1}}^j, \mathbf{y}_{k+1}). \tag{2}$$

The BPF and FA-APF correspond respectively to having

$$W_{k+1}^{j,\mathrm{BPF}} = g(\widehat{\mathbf{x}}_{t_{k+1}}^j, \mathbf{y}_{k+1}) \qquad \text{and} \qquad W_{k+1}^{j,\mathrm{FA\text{-}APF}} = \mathbb{E}[g(\mathbf{X}_{t_{k+1}}, \mathbf{y}_{k+1}) \mid \mathbf{X}_{t_k} = \mathbf{x}_{t_k}^j]. \tag{3}$$

The weights are such that $\pi_{k+1}(d\mathbf{x}) \approx \sum_{j=1}^{M} \overline{W}_{k+1}^j \delta(d\mathbf{x}; \mathbf{x}_{t_{k+1}}^j)$. The *resampling* step consists in defining a new set of particles $\mathbf{x}_{t_{k+1}}^{1:M}$ with $\mathbb{P}(\mathbf{x}_{t_{k+1}}^j = \widehat{\mathbf{x}}_{t_{k+1}}^i) = \overline{W}_{k+1}^i$. This resampling scheme ensures that the equally weighted set of particles $\mathbf{x}_{t_{k+1}}^{1:M}$ provides a Monte Carlo approximation of the

filtering distribution at time $t_{k+1}$ in the sense that $\pi_{k+1}(d\mathbf{x}) \approx M^{-1} \sum_{j=1}^{M} \delta(d\mathbf{x}; \mathbf{x}_{t_{k+1}}^{j})$. Note that the particles $\mathbf{x}_{t_{k+1}}^{1:M}$ do not need to be resampled independently given the set of propagated particles $\widehat{\mathbf{x}}_{t_{k+1}}^{1:M}$. We refer the reader to Gerber et al. (2019) for a recent discussion of resampling schemes within PFs and to Del Moral (2004) for a book-length treatment of the convergence properties of this class of Monte Carlo methods.

In most settings, the FA-APF (Pitt & Shephard, 1999) that minimizes a local variance criterion (Doucet et al., 2009) generates particles that are more consistent with informative data and weights that exhibit significantly less variability compared to the BPF. This gain in efficiency can be very substantial when the signal-to-noise ratio is high or when observations contain outliers under the model specification. Nevertheless, implementing FA-APF requires sampling from the transition probability $q_{k+1}^{\text{FA-APF}}(d\widehat{\mathbf{x}} \mid \mathbf{x})$, which is typically not feasible in practice. We will show in the following that this can be achieved in our setting by simulating a conditioned diffusion.

### 2.3 CONDITIONED AND CONTROLLED DIFFUSIONS

As the diffusion process (1) is assumed to be time-homogeneous, it suffices to focus on the initial interval $[0, T]$ and study the dynamics of the diffusion $\mathbf{X}_{[0,T]} = \{\mathbf{X}_t\}_{t \in [0,T]}$ conditioned upon the first observation $\mathbf{Y}_T = \mathbf{y}$. It is a standard result that the conditioned diffusion is described by diffusion process with the same volatility as the original diffusion but with a time-dependent drift function that takes the future observation $\mathbf{Y}_T = \mathbf{y}$ into account.

Before deriving the exact form of the conditioned diffusion, the notion of controlled diffusion needs to be discussed. For an arbitrary *control* function $\mathbf{c} : \mathcal{X} \times \mathcal{Y} \times [0, T] \to \mathbb{R}^d$ and $\mathbf{y} \in \mathcal{Y}$, consider the controlled diffusion $\{\mathbf{X}_t^{\mathbf{c},\mathbf{y}}\}_{t \in [0,T]}$ with generator $\mathcal{L}^{\mathbf{c},\mathbf{y},t}\varphi(\mathbf{x}) = \mathcal{L}\varphi(\mathbf{x}) + \langle [\sigma\mathbf{c}](\mathbf{x}, \mathbf{y}, t), \nabla\varphi(\mathbf{x}) \rangle$ and dynamics

$$d\mathbf{X}_t^{\mathbf{c},\mathbf{y}} = \underbrace{\mu(\mathbf{X}_t^{\mathbf{c},\mathbf{y}})\, dt + \sigma(\mathbf{X}_t^{\mathbf{c},\mathbf{y}})\, d\mathbf{B}_t}_{\text{(original dynamics)}} + \underbrace{[\sigma\, \mathbf{c}](\mathbf{X}_t^{\mathbf{c},\mathbf{y}}, \mathbf{y}, t)\, dt}_{\text{(control drift term)}}. \tag{4}$$

If $\mathbb{P}_{[0,T]}$ and $\mathbb{P}_{[0,T]}^{\mathbf{c},\mathbf{y}}$ denote the probability measures on the space of continuous functions $C([0, T], \mathbb{R}^d)$ generated by the original and controlled diffusions, Girsanov's theorem shows that

$$\frac{d\mathbb{P}_{[0,T]}}{d\mathbb{P}_{[0,T]}^{\mathbf{c},\mathbf{y}}}(\mathbf{X}_{[0,T]}) = \exp\left\{ -\frac{1}{2}\int_0^T \|\mathbf{c}(\mathbf{X}_t, \mathbf{y}, t)\|^2\, dt - \int_0^T \langle \mathbf{c}(\mathbf{X}_t, \mathbf{y}, t), d\mathbf{B}_t \rangle \right\}. \tag{5}$$

We now describe the optimal control function $\mathbf{c}_\star : \mathcal{X} \times \mathcal{Y} \times [0, T] \to \mathbb{R}^d$ that is such that, for any observation value $\mathbf{y} \in \mathcal{Y}$, the controlled diffusion $\mathbf{X}_{[0,T]}^{\mathbf{c}_\star,\mathbf{y}}$ has the same dynamics as the original diffusion $\mathbf{X}_{[0,T]}$ conditioned upon the observation $\mathbf{Y}_T = \mathbf{y}$. For this purpose, consider the function

$$h(\mathbf{x}, \mathbf{y}, t) = \mathbb{E}[g(\mathbf{X}_T, \mathbf{y}) \mid \mathbf{X}_t = \mathbf{x}] = \int_{\mathcal{X}} g(\mathbf{x}_T, \mathbf{y})\, p_{T-t}(d\mathbf{x}_T \mid \mathbf{x}) \tag{6}$$

that gives the probability of observing $\mathbf{Y}_T = \mathbf{y}$ when the diffusion has state $\mathbf{x} \in \mathcal{X}$ at time $t \in [0, T]$. Recall that the likelihood function $g : \mathcal{X} \times \mathcal{Y} \to \mathbb{R}_+$ was defined in Section 2.1. Equation (6) implies that $h : \mathcal{X} \times \mathcal{Y} \times [0, T] \to \mathbb{R}_+$ satisfies the backward Kolmogorov equation (Oksendal, 2013),

$$(\partial_t + \mathcal{L})h = 0, \tag{7}$$

with terminal condition $h(\mathbf{x}, \mathbf{y}, T) = g(\mathbf{x}, \mathbf{y})$ for all $(\mathbf{x}, \mathbf{y}) \in \mathcal{X} \times \mathcal{Y}$. As described in Appendix A.1, the theory of Doob's $h$-transformed shows that the optimal control is given by

$$\mathbf{c}_\star(\mathbf{x}, \mathbf{y}, t) = [\sigma^\top \nabla \log h](\mathbf{x}, \mathbf{y}, t). \tag{8}$$

We refer readers to Rogers & Williams (2000) for a formal treatment of Doob's $h$-transform.

## 3 METHOD

### 3.1 NONLINEAR FEYNMAN-KAC FORMULA

Obtaining the control function $\mathbf{c}_\star(\mathbf{x}, \mathbf{y}, t) = [\sigma^\top \nabla \log h](\mathbf{x}, \mathbf{y}, t)$ by solving the backward Kolmogorov equation in (7) for each observation $\mathbf{y} \in \mathcal{Y}$ is computationally not feasible when filtering

many observations. Furthermore, when the dimensionality of the state-space $\mathcal{X}$ becomes larger, standard numerical methods for solving Partial Differential Equations (PDEs) such as Finite Differences or the Finite Element Method become impractical. For these reasons, we propose instead to approximate the control function $\mathbf{c}_\star$ with neural networks, and employ methods based on automatic differentiation and the nonlinear Feynman-Kac approach to solve semilinear PDEs (Hartmann et al., 2017; 2019; Kebiri et al., 2017; E et al., 2017; Chan-Wai-Nam et al., 2019; Hutzenthaler & Kruse, 2020; Hutzenthaler et al., 2020; Beck et al., 2019; Han et al., 2018; Nüsken & Richter, 2021).

As the non-negative function $h$ typically decays exponentially for large $\|\mathbf{x}\|$, it is computationally more stable to work on the logarithmic scale and approximate the *value* function $v(\mathbf{x}, \mathbf{y}, t) = -\log[h(\mathbf{x}, \mathbf{y}, t)]$. Using the fact that $h$ satisfies the PDE (7), the value function satisfies

$$(\partial_t + \mathcal{L})v = \frac{1}{2}\|\sigma^\top \nabla v\|^2, \quad v(\mathbf{x}, \mathbf{y}, T) = -\log[g(\mathbf{x}, \mathbf{y})] \quad \text{for all} \quad (\mathbf{x}, \mathbf{y}) \in \mathcal{X} \times \mathcal{Y}. \quad (9)$$

Let $\{\mathbf{X}_t^{\mathbf{c}, \mathbf{y}}\}_{t \in [0,T]}$ be a controlled diffusion defined in Equation (4) for a given control function $\mathbf{c} : \mathcal{X} \times \mathcal{Y} \times [0, T] \to \mathbb{R}^d$ and define the diffusion process $\{V_t\}_{t \in [0,T]}$ as $V_t = v(\mathbf{X}_t^{\mathbf{c}, \mathbf{y}}, \mathbf{y}, t)$. While any control function $\mathbf{c}(\mathbf{x}, \mathbf{y}, t)$ with mild growth and regularity assumptions can be considered within our framework, we will see that iterative schemes that choose it as a current approximation of $\mathbf{c}_\star(\mathbf{x}, \mathbf{y}, t)$ tend to perform better in practice. Since we have that $\partial_t v + \mathcal{L}v + \langle \sigma \mathbf{c}, \nabla v \rangle = (1/2)\|\sigma^\top \nabla v\|^2 + \langle \mathbf{c}, \sigma^\top \nabla v \rangle$, Itô's Lemma shows that for any observation $\mathbf{Y}_T = \mathbf{y}$ and $0 \le s \le T$, we have

$$V_T = V_s + \int_s^T \left( \frac{1}{2}\|\mathbf{Z}_t\|^2 + \langle \mathbf{c}, \mathbf{Z}_t \rangle \right) dt + \int_s^T \langle \mathbf{Z}_t, d\mathbf{B}_t \rangle$$

with $\mathbf{Z}_t = [\sigma^\top \nabla v](\mathbf{X}_t^{\mathbf{c}, \mathbf{y}}, \mathbf{y}, t)$ and $V_T = -\log[g(\mathbf{X}_T^{\mathbf{c}, \mathbf{y}}, \mathbf{y})]$. For notational simplicity, we suppressed the dependence of $(V_t, \mathbf{Z}_t)$ on the control $\mathbf{c}$ and observation $\mathbf{y}$. In summary, the pair of processes $(V_t, \mathbf{Z}_t)$ defined as $V_t = v(\mathbf{X}_t^{\mathbf{c}, \mathbf{y}}, \mathbf{y}, t)$ and $\mathbf{Z}_t = [\sigma^\top \nabla v](\mathbf{X}_t^{\mathbf{c}, \mathbf{y}}, \mathbf{y}, t)$ are such that the following equation holds,

$$-\log[g(\mathbf{X}_T^{\mathbf{c}, \mathbf{y}}, \mathbf{y})] = V_s + \int_s^T \left\{ \frac{1}{2}\|\mathbf{Z}_t\|^2 + \langle \mathbf{c}, \mathbf{Z}_t \rangle \right\} dt + \int_s^T \langle \mathbf{Z}_t, d\mathbf{B}_t \rangle. \quad (10)$$

Crucially, under mild growth and regularity assumptions on the drift and volatility function $\mu : \mathcal{X} \to \mathbb{R}^d$ and $\sigma : \mathcal{X} \to \mathbb{R}^{d,d}$, the pair of processes $(V_t, \mathbf{Z}_t)$ is the unique solution to Equation (10) (Pardoux & Peng, 1990; 1992; Pardoux & Tang, 1999; Yong & Zhou, 1999). This result can be used as a building block for designing Monte Carlo approximations of the solution to semilinear and fully nonlinear PDEs (E et al., 2017; Han et al., 2018; Raissi, 2018; Beck et al., 2019; Huré et al., 2020; Pham et al., 2021).

## 3.2 COMPUTATIONAL DOOB'S $h$-TRANSFORM

As before, consider a diffusion $\{\mathbf{X}_t^{\mathbf{c}, \mathbf{y}}\}_{t \in [0,T]}$ controlled by a function $\mathbf{c} : \mathcal{X} \times \mathcal{Y} \times [0, T] \to \mathbb{R}^d$ and driven by the standard Brownian motion $\{\mathbf{B}_t\}_{t \ge 0}$. Furthermore, for two functions $N_0 : \mathcal{X} \times \mathcal{Y} \to \mathbb{R}$ and $N : \mathcal{X} \times \mathcal{Y} \times [0, T] \to \mathbb{R}^d$, consider the diffusion process $\{V_t\}_{t \in [0,T]}$ defined as

$$V_s = V_0 + \int_0^s \left\{ \frac{1}{2}\|\mathbf{Z}_t\|^2 + \langle \mathbf{c}(\mathbf{X}_t^{\mathbf{c}, \mathbf{y}}, \mathbf{y}, t), \mathbf{Z}_t \rangle \right\} dt + \int_0^s \langle \mathbf{Z}_t, d\mathbf{B}_t \rangle, \quad (11)$$

where the initial condition $V_0$ and the process $\{\mathbf{Z}_t\}_{t \in [0,T]}$ are defined as

$$V_0 = N_0(\mathbf{X}_0^{\mathbf{c}, \mathbf{y}}, \mathbf{y}) \qquad \text{and} \qquad \mathbf{Z}_t = N(\mathbf{X}_t^{\mathbf{c}, \mathbf{y}}, \mathbf{y}, t). \quad (12)$$

Importantly, we remind the reader that the two diffusion processes $\mathbf{X}_t^{\mathbf{c}, \mathbf{y}}$ and $V_t$ are driven by the same Brownian motion $\mathbf{B}_t$. The uniqueness result mentioned at the end of Section 3.1 implies that, if for any choice of initial condition $\mathbf{X}_0^{\mathbf{c}, \mathbf{y}} \in \mathcal{X}$ and terminal observation $\mathbf{y} \in \mathcal{Y}$ the condition $V_T = -\log[g(\mathbf{X}_T^{\mathbf{c}, \mathbf{y}}, \mathbf{y})]$ is satisfied, then we have that for all $(\mathbf{x}, \mathbf{y}, t) \in \mathcal{X} \times \mathcal{Y} \times [0, T]$

$$N_0(\mathbf{x}, \mathbf{y}) = -\log h(\mathbf{x}, \mathbf{y}, 0) \qquad \text{and} \qquad N(\mathbf{x}, \mathbf{y}, t) = -[\sigma^\top \nabla \log h](\mathbf{x}, \mathbf{y}, t). \quad (13)$$

In particular, the optimal control is given by $\mathbf{c}_\star(\mathbf{x}, \mathbf{y}, t) = -N(\mathbf{x}, \mathbf{y}, t)$. These remarks suggest parametrizing the functions $N_0(\cdot, \cdot)$ and $N(\cdot, \cdot, \cdot)$ by two neural networks with respective parameters $\theta_0 \in \Theta_0$ and $\theta \in \Theta$ while minimizing the loss function

$$\mathcal{L}(\theta_0, \theta; \mathbf{c}) = \mathbb{E}\left[ \left( V_T + \log[g(\mathbf{X}_T^{\mathbf{c}, \mathbf{Y}}, \mathbf{Y})] \right)^2 \right]. \quad (14)$$

The above expectation is with respect to the Brownian motion $\{\mathbf{B}_t\}_{t\geq 0}$, the initial condition $\mathbf{X}_0^{\mathbf{c},\mathbf{Y}} \sim \eta_{\mathbf{X}}(d\mathbf{x})$ of the controlled diffusion, and the observation $\mathbf{Y} \sim \eta_{\mathbf{Y}}(d\mathbf{y})$ at time $T$. In (14), we fix the dynamics of $\mathbf{X}_t^{\mathbf{c},\mathbf{y}}$ and optimize over the dynamics of $V_t$. The spread of the distributions $\eta_{\mathbf{X}}$ and $\eta_{\mathbf{Y}}$ should be large enough to cover typical states under the filtering distributions $\pi_k, k \geq 1$ and future observations to be filtered respectively. Specific choices will be detailed for each application in Section 5. For offline problems, one could learn in a data-driven manner by selecting $\eta_{\mathbf{Y}}$ as the empirical distribution of actual observations. We stress that these choices only impact training of the neural networks, and will not affect the asymptotic guarantees of our filtering approximations.

**CDT algorithm.** The following outlines our training procedure to learn neural networks $N_0$ and $N$ that satisfy (13). To minimize the loss function (14), any stochastic gradient algorithm can be used with a user-specified *mini-batch* size of $J \geq 1$. The following steps are iterated until convergence.

1. Choose a control $\mathbf{c} : \mathcal{X} \times \mathcal{Y} \times [0, T] \to \mathbb{R}^d$, possibly based on the current neural network parameters $(\theta_0, \theta) \in \Theta_0 \times \Theta$.

2. Simulate independent Brownian paths $\mathbf{B}_{[0,T]}^j$, initial conditions $\mathbf{X}_0^j \sim \eta_{\mathbf{X}}(d\mathbf{x})$, and observations $\mathbf{Y}^j \sim \eta_{\mathbf{Y}}(d\mathbf{y})$ for $1 \leq j \leq J$.

3. Generate the controlled trajectories: the $j$-th sample path $\mathbf{X}_{[0,T]}^j$ is obtained by forward integration of the controlled dynamics in Equation (4) with initial condition $\mathbf{X}_0^j$, control $\mathbf{c}(\cdot, \mathbf{Y}^j, \cdot)$, and the Brownian path $\mathbf{B}_{[0,T]}^j$.

4. Generate the value trajectories: the $j$-th sample path $V_{[0,T]}^j$ is obtained by forward integration of the dynamics in Equation (11)–(12) with the Brownian path $\mathbf{B}_{[0,T]}^j$ and the current neural network parameters $(\theta_0, \theta) \in \Theta_0 \times \Theta$.

5. Construct a Monte Carlo estimate of the loss function (14):

$$\widehat{\mathcal{L}} = J^{-1} \sum_{j=1}^{J} (V_T^j + \log[g(\mathbf{X}_T^j, \mathbf{Y}^j)])^2 \tag{15}$$

6. Use automatic differentiation to compute $\partial_{\theta_0}\widehat{\mathcal{L}}$ and $\partial_{\theta}\widehat{\mathcal{L}}$ and update the parameters $(\theta_0, \theta)$.

Importantly, if the control function $\mathbf{c}$ in *Step:1* does depend on the current parameters $(\theta_0, \theta)$, the gradient operations executed in *Step:6* should not be propagated through the control function $\mathbf{c}$. A standard `stop-gradient` operation available in most popular automatic differentiation frameworks can be used for this purpose.

**Time-discretization of diffusions.** For clarity of exposition, we have described our algorithm in continuous-time. In practice, one would have to discretize these diffusion processes, which is entirely straightforward. Although any numerical integrator could potentially be considered, the experiments in Section 5 employed the standard Euler–Maruyama scheme (Kloeden & Platen, 1992).

**Parametrizations of functions $N_0$ and $N$.** In all numerical experiments presented in Section 5, the functions $N_0$ and $N$ are parametrized with fully-connected neural networks with two hidden layers, number of neurons that grow linearly with dimension $d$, and the Leaky ReLU activation function except in the last layer. Future work could explore other neural network architectures for our setting. In situations that are close to a Gaussian setting (e.g. Ornstein–Uhlenbeck process observed with additive Gaussian noise) where the value function has the form $v(\mathbf{x}, \mathbf{y}, t) = \langle \mathbf{x}, a(\mathbf{y}, t)\mathbf{x} \rangle + \langle b(\mathbf{y}, t), \mathbf{x} \rangle + c(\mathbf{y}, t)$, a more parsimonious parametrization could certainly be exploited. Furthermore, the function $N(\mathbf{x}, \mathbf{y}, t)$ could be parametrized to automatically satisfy the terminal condition $N(\mathbf{x}, \mathbf{y}, T) = -[\sigma^\top \nabla \log g](\mathbf{x}, \mathbf{y})$. A possible approach consists in setting $N(\mathbf{x}, \mathbf{y}, t) = (1 - t/T)\widetilde{N}(\mathbf{x}, \mathbf{y}, t) - (t/T)[\sigma^\top \nabla \log g](\mathbf{x}, \mathbf{y})$ for some neural network $\widetilde{N} : \mathcal{X} \times \mathcal{Y} \times [0, T] \to \mathbb{R}^d$. These strategies have not be used in the experiments of Section 5.

**Choice of controlled dynamics.** In challenging scenarios where observations are highly informative and/or extreme under the model, choosing a good control function to implement *Step:1* of the proposed algorithm can be crucial. We focus on two possible implementations:

- **CDT static scheme:** a simple (and naive) choice is not using any control, i.e. $\mathbf{c}(\mathbf{x}, \mathbf{y}, t) \equiv 0 \in \mathbb{R}^d$ for all $(\mathbf{x}, \mathbf{y}, t) \in \mathcal{X} \times \mathcal{Y} \times [0, T]$.
- **CDT iterative scheme:** use the current approximation of the optimal control $\mathbf{c}_\star$ described by the parameters $(\theta_0, \theta) \in \Theta_0 \times \Theta$. This corresponds to setting $\mathbf{c}(\mathbf{x}, \mathbf{y}, t) = -N(\mathbf{x}, \mathbf{y}, t)$.

While using a *static control* approach can perform reasonably well in some situations, our results in Section 5 suggest that the *iterative control* procedure is a more reliable strategy. This is consistent with findings in the stochastic optimal control literature (Thijssen & Kappen, 2015; Pereira et al., 2019). This choice of control function drives the forward process $\mathbf{X}_t^{\mathbf{c}, \mathbf{y}}$ to regions of the state-space where the likelihood function is large and helps mitigate convergence and stability issues. Furthermore, Section 5 reports that (at convergence), the solutions $N_0$ and $N$ can be significantly different. The *iterative control* procedure leads to more accurate solutions and, ultimately, better performance when used for online filtering.

## 3.3 ONLINE FILTERING

Before performing online filtering, we first run the CDT algorithm described in Section 3.2 to construct an approximation of the optimal control $\mathbf{c}_\star(\mathbf{x}, \mathbf{y}, t) = [\sigma^\top \nabla \log h](\mathbf{x}, \mathbf{y}, t)$. For concreteness, denote by $\widehat{\mathbf{c}} : \mathcal{X} \times \mathcal{Y} \times [0, T] \to \mathbb{R}^d$ the resulting approximate control, i.e. $\widehat{\mathbf{c}}(\mathbf{x}, \mathbf{y}, t) = -N(\mathbf{x}, \mathbf{y}, t)$ where $N(\cdot, \cdot, \cdot)$ is parametrized by the final parameter $\theta \in \Theta$. Similarly, denote by $\widehat{V}_0 : \mathcal{X} \times \mathcal{Y} \to \mathbb{R}$ the approximation of the initial value function $v(\mathbf{x}, \mathbf{y}, 0) = -\log h(\mathbf{x}, \mathbf{y}, 0)$, i.e. $\widehat{V}_0(\mathbf{x}, \mathbf{y}) = N_0(\mathbf{x}, \mathbf{y})$ where $N_0(\cdot, \cdot)$ is parametrized by the final parameter $\theta_0 \in \Theta_0$.

For implementing online filtering with $M \geq 1$ particles, consider a current approximation $\pi_k(d\mathbf{x}) = M^{-1} \sum_{j=1}^M \delta(d\mathbf{x}; \mathbf{x}_{t_k}^j)$ of the filtering distribution at time $t_k \geq 0$. Given the future observation $\mathbf{Y}_{k+1} = \mathbf{y}_{k+1}$, the particles $\mathbf{x}_{t_k}^{1:M}$ are then propagated forward by exploiting the approximately optimal control $(\mathbf{x}, t) \mapsto \widehat{\mathbf{c}}(\mathbf{x}, \mathbf{y}_{k+1}, t - t_k)$. In particular, $\widehat{\mathbf{x}}_{t_{k+1}}^j$ is obtained by setting $\widehat{\mathbf{x}}_{t_{k+1}}^j = \widehat{\mathbf{X}}_{t_{k+1}}^j$ where $\{\widehat{\mathbf{X}}_t^j\}_{t \in [t_k, t_{k+1}]}$ follows the controlled diffusion

$$d\widehat{\mathbf{X}}_t^j = \underbrace{\mu(\widehat{\mathbf{X}}_t^j)\, dt + \sigma(\widehat{\mathbf{X}}_t^j)\, d\mathbf{B}_t^j}_{\text{(original dynamics)}} + \underbrace{[\sigma\widehat{\mathbf{c}}](\widehat{\mathbf{X}}_t^j, \mathbf{y}_{k+1}, t - t_k)\, dt}_{\text{(approximately optimal control)}} \tag{16}$$

initialized at $\widehat{\mathbf{X}}_{t_k}^j = \mathbf{x}_{t_k}^j$. Each propagated particle $\widehat{\mathbf{x}}_{t_{k+1}}^j$ is associated with a normalized weight $\overline{W}_{k+1}^j = W_{k+1}^j / \sum_{i=1}^M W_{k+1}^i$ where $W_{k+1}^j = (d\mathbb{P}_{[t_k, t_{k+1}]} / d\mathbb{P}_{[t_k, t_{k+1}]}^{\widehat{\mathbf{c}}, \mathbf{y}_{k+1}})(\widehat{\mathbf{X}}_{[t_k, t_{k+1}]}^j) \times g(\widehat{\mathbf{x}}_{t_{k+1}}^j, \mathbf{y}_{k+1})$. We recall that the probability measures $\mathbb{P}_{[t_k, t_{k+1}]}$ and $\mathbb{P}_{[t_k, t_{k+1}]}^{\widehat{\mathbf{c}}, \mathbf{y}_{k+1}}$ correspond to the original and controlled diffusions on the interval $[t_k, t_{k+1}]$. Girsanov's theorem, as described in Equation (5), implies that

$$W_{k+1}^j = \exp\left\{ -\frac{1}{2} \int_{t_k}^{t_{k+1}} \|\mathbf{Z}_t^j\|^2\, dt + \int_{t_k}^{t_{k+1}} \langle \mathbf{Z}_t^j, d\mathbf{B}_t^j \rangle + \log g(\mathbf{x}_{t_{k+1}}^j, \mathbf{y}_{k+1}) \right\}$$

where $\mathbf{Z}_t^j = -\widehat{\mathbf{c}}(\widehat{\mathbf{X}}_t^j, \mathbf{y}_{k+1}, t - t_k)$. Similarly to Equation (11), consider the diffusion process $\{V_t^j\}_{t \in [t_k, t_{k+1}]}$ defined by the dynamics $dV_t^j = -\frac{1}{2}\|\mathbf{Z}_t^j\|^2\, dt + \langle \mathbf{Z}_t^j, d\mathbf{B}_t^j \rangle$ with initialization at $V_{t_k}^j = \widehat{V}_0(\mathbf{x}_{t_k}^j, \mathbf{y}_{k+1})$. Therefore the weight can be re-written as

$$W_{k+1}^j = \exp\left\{ \underbrace{V_{t_{k+1}}^j + \log g(\mathbf{x}_{t_{k+1}}^j, \mathbf{y}_{k+1})}_{\approx 0} \right\} \exp\left\{ -\widehat{V}_0(\mathbf{x}_{t_k}^j, \mathbf{y}_{k+1}) \right\}, \tag{17}$$

and computed by numerically integrating the process $\{V_t^j\}_{t \in [t_k, t_{k+1}]}$. Given the definition of the loss function in (14), we can expect the term within the first exponential to be close to zero. In the ideal case where $\widehat{\mathbf{c}}(\mathbf{x}, \mathbf{y}, t) \equiv \mathbf{c}_\star(\mathbf{x}, \mathbf{y}, t)$ and $\widehat{V}_0(\mathbf{x}, \mathbf{y}) \equiv -\log h(\mathbf{x}, \mathbf{y}, 0)$, one recovers the exact AF-APF weights in (3). Once the unnormalized weights (17) are computed, the resampling steps are identical to those described in Section 2.2 for a standard PF. For practical implementations, all the processes involved in the proposed methodology can be straightforwardly time-discretized. To distinguish between CDT learning with static or iterative control, we shall refer to the resulting approximation of FA-APF as Static-APF and Iterative-APF respectively. We note that these APFs do not involve modified resampling probabilities as described e.g. in Chopin et al. (2020, p. 145).

## 4 RELATED WORK

This section positions our work within the existing literature.

**MCMC methods:** Several works have developed MCMC methods for smoothing and parameter estimation of SDEs; for example, Roberts & Stramer (2001) proposes to treat paths between observations as missing data. Our work concentrates on the online filtering problem: this cannot be tackled with MCMC methods.

**Exact Simulation:** Several methods have been proposed to reduce or eliminate the bias due to discretization (Beskos et al., 2006a;b; Fearnhead et al., 2010; 2008); these methods typically rely on the Lamperti transform that is only rarely available in multivariate settings. Furthermore, when filtering diffusion with highly-informative observations, the discretization bias is often orders of magnitude smaller than other sources of errors. We also stress that our method is generic: it does not exploit any specific structure of the diffusion process being assimilated.

**Gaussian Assumptions:** In the data-assimilation literature, methods based on variations of the Ensemble Kalman Filter (EnKF) Evensen (2003) have been successfully deployed in applied scenarios and very high-dimensional settings. These methods do rely on strong Gaussian assumptions and are inappropriate for highly nonlinear and non-Gaussian models. In contrast, our method is asymptotically exact in the limit when the number of particles $M \to \infty$ (up to discretization error). Indeed, we do not expect our method to be competitive with this class of (approximate) methods in very high-dimensional settings that are common in numerical weather forecasting. These methods typically achieve lower variance by increasing the bias. Our method is designed to filter diffusion processes in low or moderate dimensional settings. It is likely that scaling our method to truly high-dimensional settings with effective dimension $D \gg 10^2$ would require introducing model-specific approximations (e.g. localization strategies).

**Steering particles towards observations:** particle methods pioneered by Van Leeuwen (2010) are based on this natural principle in order to mitigate collapse of PFs in high-dimensional settings. These methods typically rely on some model structure (e.g. linear Gaussian observation process) and have a number of tuning parameters. They can be understood as parametrizing a linear control, which is only expected to work well for linear Gaussian dynamics, admittedly very important in applications such as geoscience.

**Implicit Particle Filter:** the method of Chorin et al. (2010) attempts to transform standard i.i.d Gaussian samples into samples from the optimal proposal density. To implement this methodology requires a number of assumptions and requires solving a non-convex optimization step for each particle and each time step. This can quickly become computational burdensome.

**Guided Intermediate Resampling Filters (GIRF):** the method of Del Moral & Murray (2015); Park & Ionides (2020) propagates particles at intermediate time intervals between observations with the original dynamics and triggers resampling steps based on a *guiding functions* that forecast the likelihood of future observations. The choice of *guiding functions* is crucial for good algorithmic performance. We note that GIRF is in fact intimately related to Doob's $h$-transform as the optimal choice of guiding functions is given by (6) (Park & Ionides, 2020). However, even under this optimal choice, the resulting GIRF is still sub-optimal when compared to an APF that moves particles using the optimal control induced by Doob's $h$-transform, i.e. it is better to move particles well rather than rely on weighting and resampling. The latter behaviour is supported by our numerical experiments. Appendix A.5 details our GIRF implementation and the connection to Doob's $h$-transform.

## 5 EXPERIMENTS

We performed numerical experiments on three different models: an Ornstein–Uhlenbeck model, a (nonlinear) Logistic diffusion model and a (nonlinear) diffusion model describing cell differentiation. This section presents experiments on the Ornstein–Uhlenbeck model; the two other studies can be found in the Appendix A.3 and A.4. All experiments employed 2000 iterations of the `Adam` optimizer with a learning rate of $0.01$ and a mini-batch size of $B = 1000$ sample paths with 10 different observations each. Appendix A.6 describes how the CDT algorithm and the approximate control functions behave during training. Training times took less than two minutes on a standard

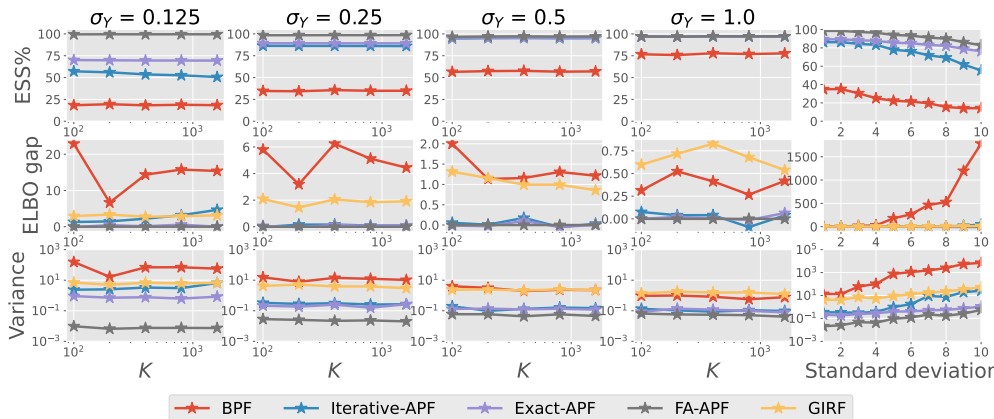

Figure 1: Results for Ornstein–Uhlenbeck model with $d = 1$ based on 100 independent repetitions of each PF. The ELBO gap in the second row is relative to FA-APF.

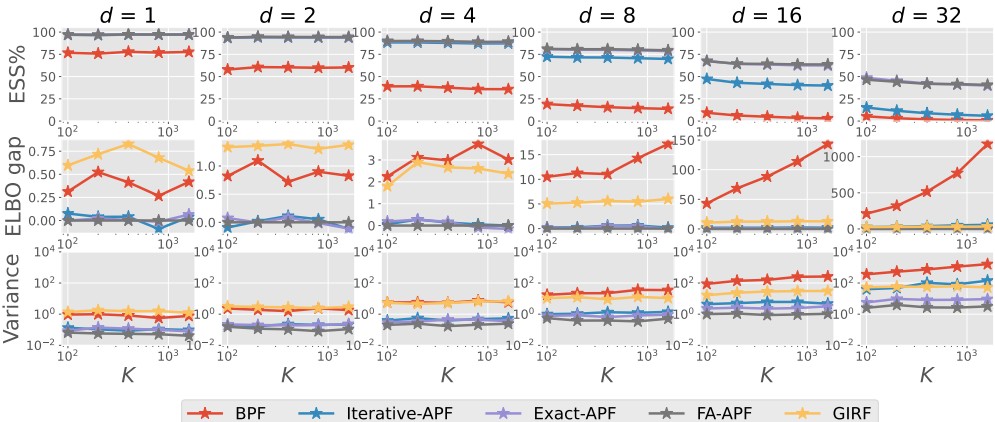

Figure 2: Results for Ornstein–Uhlenbeck model with $\sigma_{\mathbf{Y}} = 1.0$ based on 100 independent repetitions of each PF. The ELBO gap in the second row is relative to FA-APF.

CPU: it is negligible when compared to the cost of running filters with many particles and/or to assimilate large number of observations. The inter-observation time was $T = 1$ and we employed the Euler–Maruyama integrator with a stepsize of $0.02$ for all examples. Our results are not sensitive to the choice of $T$ and discretization stepsize as long as it is sufficiently small. We report the Effective Sample Size (ESS) averaged over observation times and independent repetitions, the evidence lower bound (ELBO) $\mathbb{E}[\log \widehat{p}(\mathbf{y}_1, \ldots, \mathbf{y}_K)]$, and the variance $\mathrm{Var}[\log \widehat{p}(\mathbf{y}_1, \ldots, \mathbf{y}_K)]$, where $\widehat{p}(\mathbf{y}_1, \ldots, \mathbf{y}_K)$ denotes its unbiased estimator of the marginal likelihood of the time-discretized filter $p(\mathbf{y}_1, \ldots, \mathbf{y}_K)$. When testing particle filters with varying number of observations $K$, we increased the number of particles $M$ linearly with $K$ to keep marginal likelihood estimators stable (Bérard et al., 2014). For non-toy models, our GIRF implementation relies on a sub-optimal but practical choice of guiding functions that gradually introduce information from the future observation by annealing the observation density using a linear (Linear-GIRF) or quadratic schedule (Quadratic-GIRF).

## 5.1 Ornstein–Uhlenbeck model

Consider a $d$-dimensional Ornstein–Uhlenbeck process given by (1) with $\mu(\mathbf{x}) = -\mathbf{x}$, $\sigma(\mathbf{x}) = \mathbf{I}_d$ and the Gaussian observation model $g(\mathbf{x}, \mathbf{y}) = \mathcal{N}(\mathbf{y}; \mathbf{x}, \sigma_{\mathbf{Y}}^2 \mathbf{I}_d)$. We chose $\eta_{\mathbf{X}} = \mathcal{N}(\mathbf{0}_d, \mathbf{I}_d/2)$ as the stationary distribution and $\eta_{\mathbf{Y}} = \mathcal{N}(\mathbf{0}_d, (1/2 + \sigma_{\mathbf{Y}}^2)\mathbf{I}_d)$ as the implied distribution of the observation when training neural networks with the CDT iterative scheme. We took different values of $\sigma_{\mathbf{Y}} \in \{0.125, 0.25, 0.5, 1.0\}$ to vary the informativeness of observations and $d \in \{1, 2, 4, 8, 16, 32\}$ to illustrate the impact of dimension. Analytical tractability in this example (Appendix A.2) consider three idealized particle filters, namely an APF with exact networks (Exact-APF), FA-APF, and GIRF with optimal guiding functions (Appendix A.5). Comparing our proposed Iterative-APF to Exact-APF and FA-APF enables us to distinguish between neural network approximation errors and time-discretization errors. We note that all PFs except the FA-APF involve time-discretization.

Columns 1 to 4 of Figure 1 summarize our numerical findings when filtering simulated observations from the model with varying $\sigma_{\mathbf{Y}}$ and fixed $d = 1$. We see that the performance of BPF deteriorates as the observations become more informative, which is to be expected. Furthermore, when $\sigma_{\mathbf{Y}}$ is small, the impact of our neural network approximation and time-discretization becomes more noticeable. For the values of $\sigma_{\mathbf{Y}}$ and the number of observations $K$ considered, Iterative-APF had substantial gains in efficiency over BPF and typically outperformed GIRF. From Column 5, we note that these gains over BPF become very large when we filter $K = 100$ observations simulated with observation standard deviations that are multiples of $\sigma_{\mathbf{Y}} = 0.25$ which was used to run the filters. In particular, while the ELBO of BPF diverges as we increase the degree of noise in the simulated observations, the ELBO of Iterative-APF and GIRF remain stable.

Figure 2 shows the impact of increasing dimension $d$ with fixed $\sigma_{\mathbf{Y}} = 1.0$ when filtering simulated observations from the model. Due to the curse of dimensionality (Snyder et al., 2008; 2015), it is not surprising for the performance of all PFs to degrade with dimension. Although the error of our neural network approximation becomes more pronounced when $d$ is large, the gain in efficiency of Iterative-APF relative to BPF is very significant in the higher dimensional regime, and particularly so when the number of observations $K$ is also large. Iterative-APF also outperformed GIRF in most settings, with comparable performance when $d$ is large.

## 6 Discussion

This paper introduced the CDT algorithm, a Sequential Monte-Carlo method for online filtering of diffusion processes evolving in state-spaces of low to moderate dimensions. Contrarily to a number of existing methods, the CDT approach is general and does not exploit any particular structure of the diffusion process. Furthermore, numerical simulations suggests that the CDT algorithm is especially worthwhile when compared to competing approaches (e.g. BPF or GIRF) in higher dimensional settings or when the observations are highly informative. Ongoing work involves extending the CDT framework to parameter estimation and experimenting with alternative formulations and/or parameterizations to accelerate the training procedures.

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

## A    APPENDIX

### A.1    DOOB'S $h$-TRANSFORM

This section gives an heuristic derivation of the Equation (8) that describes the optimal control. To simplify notation, we shall denote the conditioned process $\mathbf{X}_{[0,T]} \mid (\mathbf{Y}_T = \mathbf{y})$ as $\widehat{\mathbf{X}}_{[0,T]}$. Recall the function

$$h(\mathbf{x}, \mathbf{y}, t) = \mathbb{E}[g(\mathbf{X}_T, \mathbf{y}) \mid \mathbf{X}_t = \mathbf{x}] = \int_{\mathcal{X}} g(\mathbf{x}_T, \mathbf{y}) \, p_{T-t}(d\mathbf{x}_T \mid \mathbf{x}) \tag{18}$$

which gives the probability of observing $\mathbf{Y}_T = \mathbf{y}$ when the diffusion process has state $\mathbf{x} \in \mathcal{X}$ at time $t \in [0, T]$. The definition in (6) implies that the function $h : \mathcal{X} \times \mathcal{Y} \times [0, T] \to \mathbb{R}_+$ satisfies the backward Kolmogorov equation (Oksendal, 2013),

$$(\partial_t + \mathcal{L})h = 0, \tag{19}$$

with terminal condition $h(\mathbf{x}, \mathbf{y}, T) = g(\mathbf{x}, \mathbf{y})$ for all $(\mathbf{x}, \mathbf{y}) \in \mathcal{X} \times \mathcal{Y}$. For $\varphi : \mathcal{X} \to \mathbb{R}$ and an infinitesimal increment $\delta > 0$, we have

$$\begin{aligned}
\mathbb{E}[\varphi(\widehat{\mathbf{X}}_{t+\delta})|\widehat{\mathbf{X}}_t = \mathbf{x}] &= \mathbb{E}[\varphi(\mathbf{X}_{t+\delta}) \, g(\mathbf{X}_T, \mathbf{y}) \mid \mathbf{X}_t = \mathbf{x}] \,/\, \mathbb{E}[g(\mathbf{X}_T, \mathbf{y})|\mathbf{X}_t = \mathbf{x}] \\
&= \mathbb{E}[\varphi(\mathbf{X}_{t+\delta}) \, h(\mathbf{X}_{t+\delta}, \mathbf{y}, t+\delta) \mid \mathbf{X}_t = \mathbf{x}] \,/\, h(\mathbf{x}, \mathbf{y}, t) \\
&= \varphi(\mathbf{x}) + \delta \left\{ \frac{\mathcal{L}[\varphi \, h]}{h} \right\}(\mathbf{x}, \mathbf{y}, t) + O(\delta^2).
\end{aligned} \tag{20}$$

Furthermore, since the function $h$ satisfies (7), some algebra shows that $\mathcal{L}[\varphi \, h]/h = \mathcal{L}\varphi + \langle \sigma\sigma^\top \nabla \log h, \nabla \varphi \rangle$. Taking $\delta \to 0$, this heuristic derivation shows that the generator of the conditioned diffusion equals $\mathcal{L}\varphi + \langle \sigma\sigma^\top \nabla \log h, \nabla \varphi \rangle$. Hence $\widehat{\mathbf{X}}_{[0,T]}$ satisfies the dynamics of a controlled diffusion (4) with control function

$$\mathbf{c}_\star(\mathbf{x}, \mathbf{y}, t) = [\sigma^\top \nabla \log h](\mathbf{x}, \mathbf{y}, t) \tag{21}$$

This proves Equation (8).

## A.2 ANALYTICAL TRACTABILITY OF THE ORNSTEIN–UHLENBECK MODEL

The transition probability of the Ornstein–Uhlenbeck process considered in Section 5.1 is

$$p_t(d\hat{\mathbf{x}} \mid \mathbf{x}) = \mathcal{N}(\hat{\mathbf{x}}; \mu_{\mathbf{X}}(\mathbf{x}, t), \sigma_{\mathbf{X}}^2(t)\mathbf{I}_d)d\hat{\mathbf{x}}$$

for time $t > 0$, with mean $\mu_{\mathbf{X}}(\mathbf{x}, t) = \mathbf{x}\exp(-t)$ and variance $\sigma_{\mathbf{X}}^2(t) = \{1 - \exp(-2t)\}/2$. From (6), we have

$$h(\mathbf{x}, \mathbf{y}, t) = \int_{\mathbb{R}^d} \mathcal{N}(\mathbf{y}; \mathbf{x}_T, \sigma_{\mathbf{Y}}^2\mathbf{I}_d)\,\mathcal{N}(\mathbf{x}_T; \mu_{\mathbf{X}}(\mathbf{x}, T-t), \sigma_{\mathbf{X}}^2(T-t)\mathbf{I}_d)d\mathbf{x}_T$$

$$= (2\pi)^{-d/2}\sigma_{\mathbf{X}}^{-d}(T-t)\sigma_{\mathbf{Y}}^{-d}\sigma_h^d(T-t)\exp\left\{\frac{1}{2}\sigma_h^2(T-t)\left\|\frac{\mu_{\mathbf{X}}(\mathbf{x}, T-t)}{\sigma_{\mathbf{X}}^2(T-t)} + \frac{\mathbf{y}}{\sigma_{\mathbf{Y}}^2}\right\|^2\right\}$$

$$\times \exp\left\{-\frac{\|\mu_{\mathbf{X}}(\mathbf{x}, T-t)\|^2}{2\sigma_{\mathbf{X}}^2(T-t)} - \frac{\|\mathbf{y}\|^2}{2\sigma_{\mathbf{Y}}^2}\right\}$$

where $\sigma_h^2(t) = \{\sigma_{\mathbf{X}}^{-2}(t) + \sigma_{\mathbf{Y}}^{-2}\}^{-1}$. Hence we can compute the value function $v(\mathbf{x}, \mathbf{y}, t) = -\log[h(\mathbf{x}, \mathbf{y}, t)]$. Next, the optimal control function is

$$\mathbf{c}_\star(\mathbf{x}, \mathbf{y}, t) = [\sigma^\top\nabla\log h](\mathbf{x}, \mathbf{y}, t)$$

$$= \frac{\sigma_h^2(T-t)\exp\{-(T-t)\}}{\sigma_{\mathbf{X}}^2(T-t)}\left\{\frac{\mu_{\mathbf{X}}(\mathbf{x}, T-t)}{\sigma_{\mathbf{X}}^2(T-t)} + \frac{\mathbf{y}}{\sigma_{\mathbf{Y}}^2}\right\} - \frac{\exp\{-(T-t)\}}{\sigma_{\mathbf{X}}^2(T-t)}\mu_{\mathbf{X}}(\mathbf{x}, T-t).$$

The distribution of $\mathbf{X}_T$ conditioned on $\mathbf{X}_0 = \mathbf{x}_0$ and $\mathbf{Y}_T = \mathbf{y}$ is $\mathcal{N}(\mu_h(\mathbf{x}_0, \mathbf{y}, T), \sigma_h^2(T)\mathbf{I}_d)$ with

$$\mu_h(\mathbf{x}_0, \mathbf{y}, T) = \sigma_h^2(T)\left\{\frac{\mu_{\mathbf{X}}(\mathbf{x}_0, T)}{\sigma_{\mathbf{X}}^2(T)} + \frac{\mathbf{y}}{\sigma_{\mathbf{Y}}^2}\right\}.$$

## A.3 LOGISTIC DIFFUSION MODEL

In this section we consider a logistic diffusion process (Dennis & Costantino, 1988; Knape & De Valpine, 2012) to model the dynamics of a population size $\{\mathbf{P}_t\}_{t\geq 0}$, defined by

$$d\mathbf{P}_t = (\theta_3^2/2 + \theta_1 - \theta_2\mathbf{P}_t)\mathbf{P}_t\,dt + \theta_3\mathbf{P}_t\,d\mathbf{B}_t. \tag{22}$$

We apply the Lamperti transformation $\mathbf{X}_t = \log(\mathbf{P}_t)/\theta_3$ and work with the process $\{\mathbf{X}_t\}_{t\geq 0}$ that satisfies (1) with $\mu(\mathbf{x}) = \theta_1/\theta_3 - (\theta_2/\theta_3)\exp(\theta_3\mathbf{x})$ and $\sigma(\mathbf{x}) = 1$. Following (Knape & De Valpine, 2012), we adopt a negative binomial observation model $g(\mathbf{x}, \mathbf{y}) = \mathcal{NB}(\mathbf{y}; \theta_4, \exp(\theta_3\mathbf{x}))$ for counts $\mathbf{y} \in \mathbb{N}_0$ with dispersion $\theta_4 > 0$ and mean $\exp(\theta_3\mathbf{x})$. We set $(\theta_1, \theta_2, \theta_3, \theta_4)$ as the parameter estimates obtained in (Knape & De Valpine, 2012). Noting that (22) admits a Gamma distribution with shape parameter $2(\theta_3^2/2 + \theta_1)/\theta_3^2 - 1$ and rate parameter $2\theta_2/\theta_3^2$ as stationary distribution (Dennis & Costantino, 1988), we select $\eta_{\mathbf{X}}$ as the push-forward under the Lamperti transformation and $\eta_{\mathbf{Y}}$ as the implied distribution of the observation when training neural networks under both static and iterative CDT schemes. To induce varying levels of informative observations, we considered $\theta_4 \in \{1.069, 4.303, 17.631, 78.161\}$.

Figure 3 displays our filtering results for various number of simulated observations from the model (Columns 1 to 4) and for $K = 100$ observations that are simulated with observation standard deviations larger than $\theta_4 = 17.631$ used to run the filters (Column 5). In the latter setup, we solved for different values of $\theta_4$ in the negative binomial observation model to induce larger standard deviations. The behaviour of BPF and Iterative-APF is similar to the previous example as the observations become more informative with larger values of $\theta_4$. Iterative-APF outperformed all other algorithms over all combinations of $\theta_4$ and $K$ considered, and also when filtering observations that are increasingly extreme under the model. We note also that the APFs trained using the CDT static scheme can sometimes give unstable results, particularly in challenging scenarios.

## A.4 CELL MODEL

This section examines a cell differentiation and development model from (Wang et al., 2011). Cellular expression levels $\mathbf{X}_t = (\mathbf{X}_{t,1}, \mathbf{X}_{t,2})$ of two genes are modelled by (1) with

$$\mu(\mathbf{x}) = \begin{pmatrix} \mathbf{x}_1^4/(2^{-4} + \mathbf{x}_1^4) + 2^{-4}/(2^{-4} + \mathbf{x}_2^4) - \mathbf{x}_1 \\ \mathbf{x}_2^4/(2^{-4} + \mathbf{x}_2^4) + 2^{-4}/(2^{-4} + \mathbf{x}_1^4) - \mathbf{x}_2 \end{pmatrix} \tag{23}$$

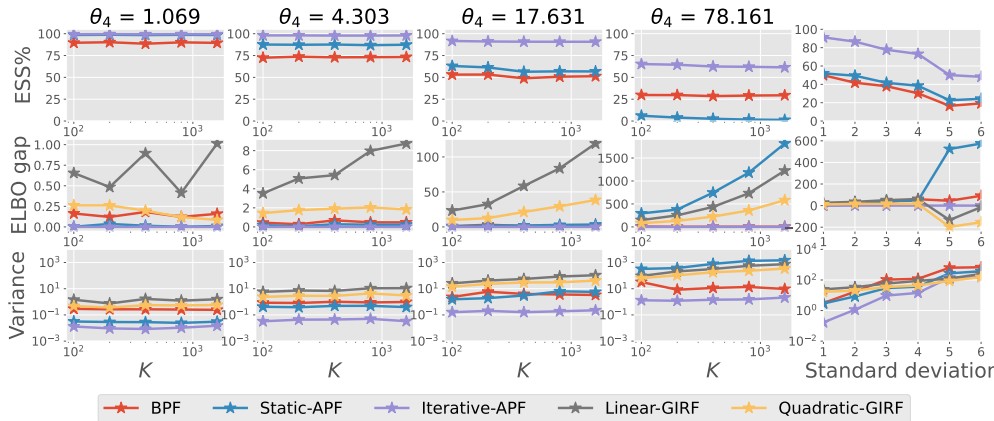

Figure 3: Results for logistic diffusion model based on 100 independent repetitions of each PF. The ELBO gap in the second row is relative to Iterative-APF.

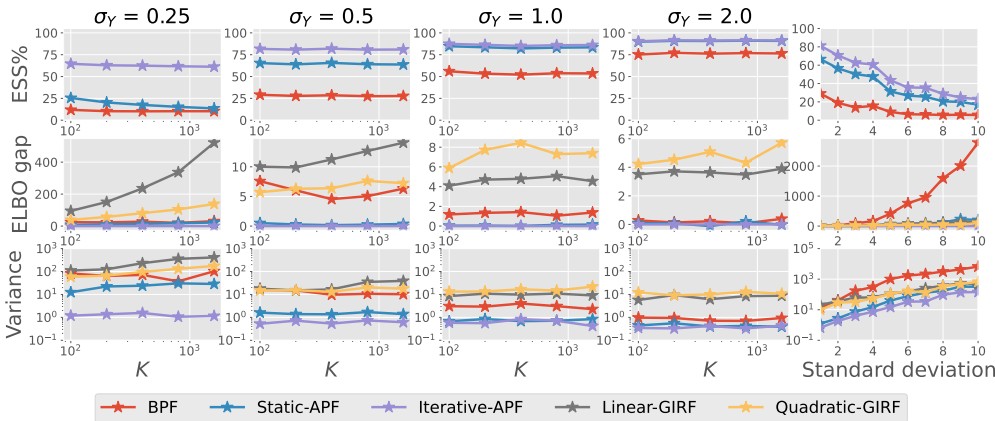

Figure 4: Results for cell model based on 100 independent repetitions of each PF. The ELBO gap in the second row is relative to Iterative-APF.

and $\sigma(\mathbf{x}) = \sqrt{0.1}\mathbf{I}_d$. The terms in (23) describe self-activation, mutual inhibition and inactivation respectively, and the volatility captures intrinsic and external fluctuations. We initialize the diffusion process from the undifferentiated state of $\mathbf{X}_0 = (1, 1)$ and consider the Gaussian observation model $g(\mathbf{x}, \mathbf{y}) = \mathcal{N}(\mathbf{y}; \mathbf{x}, \sigma_{\mathbf{Y}}^2 \mathbf{I}_2)$. To train neural networks under both static and iterative CDT schemes, we selected $\eta_{\mathbf{X}}$ and $\eta_{\mathbf{Y}}$ as the empirical distributions obtained by simulating states and observations from the model for 2000 time units.

Figure 4 illustrates our numerical results for various number of observations $K$ and $\sigma_{\mathbf{Y}} \in \{0.25, 0.5, 1.0, 2.0\}$. It shows that Iterative-APF offers significant gains over all other algorithms when filtering observations that are informative (see Columns 1 to 4) and highly extreme under the model specification of $\sigma_{\mathbf{Y}} = 0.5$ (see Column 5). In this example, Static-APF did not exhibit any unstable behaviour and its performance lies somewhere in between BPF and Iterative-APF.

## A.5 GUIDED INTERMEDIATE RESAMPLING FILTERS.

We first describe our implementation of GIRF for online filtering. For $M \geq 1$ particles, let $\pi_k(d\mathbf{x}) = M^{-1} \sum_{j=1}^M \delta(d\mathbf{x}; \mathbf{x}_{t_k}^j)$ denote a current approximation of the filtering distribution at time $t_k \geq 0$. Given the future observation $\mathbf{Y}_{k+1} = \mathbf{y}_{k+1}$ at time $t_{k+1}$, GIRF introduces a sequence of intermediate time steps $t_k = s_0 < s_1 < \cdots < s_P = t_{k+1}$ between the observation times, and a

sequence of guiding functions $\{G_p\}_{p=0}^P$ satisfying

$$G_0(\mathbf{x}_{s_0}, \mathbf{y}_{k+1}) \prod_{p=1}^P G_p(\mathbf{x}_{s_{p-1}}, \mathbf{x}_{s_p}, \mathbf{y}_{k+1}) = g(\mathbf{x}_{t_{k+1}}, \mathbf{y}_{k+1}). \tag{24}$$

For each intermediate step $p \in \{1, \ldots, P\}$, the particles $\mathbf{x}_{s_p}^{1:M}$ are then propagated forward according to the original SDE (1), i.e. $\widehat{\mathbf{x}}_{s_{p+1}}^j \sim p_{\Delta s_{p+1}}(d\widehat{\mathbf{x}} \mid \mathbf{x}_{s_p}^j)$ with stepsize $\Delta s_{p+1} = s_{p+1} - s_p$. In practice, this propagation step can be replaced by a numerical integrator. Each particle $\widehat{\mathbf{x}}_{s_{p+1}}^j$ is then associated with a normalized weight $\overline{W}_{p+1}^j = W_{p+1}^j / \sum_{i=1}^M W_{p+1}^i$, where the unnormalized weight

$$W_p^j = G_p(\mathbf{x}_{s_{p-1}}^j, \widehat{\mathbf{x}}_{s_p}^j, \mathbf{y}_{k+1}), \quad p \in \{1, \ldots, P-1\},$$
$$W_P^j = G_P(\mathbf{x}_{s_{P-1}}^j, \widehat{\mathbf{x}}_{s_P}^j, \mathbf{y}_{k+1}) G_0(\widehat{\mathbf{x}}_{s_P}^j, \mathbf{y}_{k+2}), \quad \text{if } t_{k+1} \text{ is not the final observation time,}$$
$$W_P^j = G_P(\mathbf{x}_{s_{P-1}}^j, \widehat{\mathbf{x}}_{s_P}^j, \mathbf{y}_{k+1}), \quad \text{if } t_{k+1} \text{ is the final observation time.}$$

After the unnormalized weights are computed, the resampling operation is the same as a standard PF (see Section 2.2).

From the above description, we see that the role of $\{G_p\}_{p=0}^P$ is to guide particles to appropriate regions of the state-space using the weighting and resampling steps. The optimal choice of guiding functions (Park & Ionides, 2020) is

$$G_0(\mathbf{x}_{s_0}, \mathbf{y}_{k+1}) = h(\mathbf{x}_{s_0}, \mathbf{y}_{k+1}, s_0), \quad G_p(\mathbf{x}_{s_{p-1}}, \mathbf{x}_{s_p}, \mathbf{y}_{k+1}) = \frac{h(\mathbf{x}_{s_p}, \mathbf{y}_{k+1}, s_p)}{h(\mathbf{x}_{s_{p-1}}, \mathbf{y}_{k+1}, s_{p-1})}, \tag{25}$$

for $p \in \{1, \ldots, P\}$, where $h : \mathcal{X} \times \mathcal{Y} \times [0, T] \to \mathbb{R}_+$ defined in (6) is given by Doob's $h$-transform. The condition (24) is satisfied as we have a telescoping product and $h(\mathbf{x}_{t_{k+1}}, \mathbf{y}_{k+1}, t_{k+1}) = g(\mathbf{x}_{t_{k+1}}, \mathbf{y}_{k+1})$. For the Ornstein–Uhlenbeck model of Section 5.1, we leveraged analytical tractability of (25) in our implementation of GIRF. When the optimal choice (25) is intractable, one sub-optimal but practice choice that gradually introduces information from the future observation by annealing the observation density is

$$G_0(\mathbf{x}_{s_0}, \mathbf{y}_{k+1}) = g(\mathbf{x}_{s_0}, \mathbf{y}_{k+1})^{\lambda_0}, \quad G_p(\mathbf{x}_{s_{p-1}}, \mathbf{x}_{s_p}, \mathbf{y}_{k+1}) = \frac{g(\mathbf{x}_{s_p}, \mathbf{y}_{k+1})^{\lambda_p}}{g(\mathbf{x}_{s_{p-1}}, \mathbf{y}_{k+1})^{\lambda_{p-1}}},$$

for $p \in \{1, \ldots, P\}$, where $\{\lambda_p\}_{p=0}^P$ is a non-decreasing sequence with $\lambda_P = 1$. This construction clearly satisfies the condition in (24). It is interesting to note that under the choice $\lambda_p = 0$ for $p \in \{1, \ldots, P-1\}$, GIRF recovers the BPF. In our numerical implementation, we considered both linear and quadratic annealing schedules $\{\lambda_p\}_{p=0}^P$ which determine the rate at which information from the future observation is introduced.

Lastly, we explain why GIRF with the optimal guiding functions (25) is still sub-optimal compared to an APF that move particles using the optimal control $\mathbf{c}_\star : \mathcal{X} \times \mathcal{Y} \times [0, T] \to \mathbb{R}^d$ induced by Doob's $h$-transform. We consider the law of $\{\mathbf{X}_{s_p}\}_{p=1}^P$ conditioned on $\mathbf{X}_{s_0} = \mathbf{x}_{s_0}$ and $\mathbf{Y}_{k+1} = \mathbf{y}_{k+1}$

$$\prod_{p=1}^P p_{\Delta s_p}(d\mathbf{x}_{s_p} \mid \mathbf{x}_{s_{p-1}}) g(\mathbf{x}_{s_P}, \mathbf{y}_{k+1}). \tag{26}$$

Under the condition (24), we can write the law (26) as

$$G_0(\mathbf{x}_{s_0}, \mathbf{y}_{k+1}) \prod_{p=1}^P p_{\Delta s_p}(d\mathbf{x}_{s_p} \mid \mathbf{x}_{s_{p-1}}) G_p(\mathbf{x}_{s_{p-1}}, \mathbf{x}_{s_p}, \mathbf{y}_{k+1}). \tag{27}$$

GIRF can be understood as a Sequential Monte Carlo (SMC) algorithm (Chopin et al., 2020) approximating the law (27) with Markov transitions $\{p_{\Delta s_p}\}_{p=1}^P$ and potential functions $\{G_p\}_{p=0}^P$ given by (25). We can rewrite (27) as

$$G_0(\mathbf{x}_{s_0}, \mathbf{y}_{k+1}) \prod_{p=1}^P p_{\Delta s_p}^h(d\mathbf{x}_{s_p} \mid \mathbf{x}_{s_{p-1}}), \tag{28}$$

where Markov transitions $\{p_{\Delta s_p}^h\}_{p=1}^P$ are defined as

$$p_{\Delta s_p}^h(d\mathbf{x}_{s_p} \mid \mathbf{x}_{s_{p-1}}) = \frac{p_{\Delta s_p}(d\mathbf{x}_{s_p} \mid \mathbf{x}_{s_{p-1}})h(\mathbf{x}_{s_p}, \mathbf{y}_{k+1}, s_p)}{h(\mathbf{x}_{s_{p-1}}, \mathbf{y}_{k+1}, s_{p-1})} \tag{29}$$

for $p \in \{1, \ldots, P\}$. By the Markov property, we have $h(\mathbf{x}_{s_{p-1}}, \mathbf{y}_{k+1}, s_{p-1}) = \int_\mathcal{X} p_{\Delta s_p}(d\mathbf{x}_{s_p} \mid \mathbf{x}_{s_{p-1}})h(\mathbf{x}_{s_p}, \mathbf{y}_{k+1}, s_p)$, hence (29) is a valid Markov transition kernel. Moreover, it follows from Dai Pra (1991, Theorem 2.1) that $\{p_{\Delta s_p}^h\}_{p=1}^P$ are the transition probabilities of the controlled diffusion process in (4) with optimal control $\mathbf{c}_\star(\mathbf{x}, \mathbf{y}, t) = [\sigma^\top \nabla \log h](\mathbf{x}, \mathbf{y}, t)$. Hence an APF propagating particles according to this optimally controlled process can be seen as SMC algorithm approximating (28) with Markov transitions $\{p_{\Delta s_p}^h\}_{p=1}^P$ and a single potential function $G_0$. By viewing GIRF and APF as specific instantaneous of SMC algorithms, it is clear that the former is sub-optimal compared to the latter. Intuitively, this means that better particle approximations can be obtained by moving particles well instead of relying on weighting and resampling.

### A.6 COMPUTATIONAL DOOB'S $h$-TRANSFORM ALGORITHM

In this section, we provide figures to illustrate how our proposed CDT algorithm behaves. We report the training curves (i.e. loss v.s. iteration), as well as describe the evolution of the approximate control functions parametrized by the neural networks. In the analytically tractable Ornstein–Uhlenbeck case, comparison with the optimal control is possible. See Figures 5 and 6 for the Ornstein–Uhlenbeck model of Section 5.1, Figures 7 and 8 for the logistic diffusion model of Section A.3, and Figure 9 for the cell model of Section A.4.

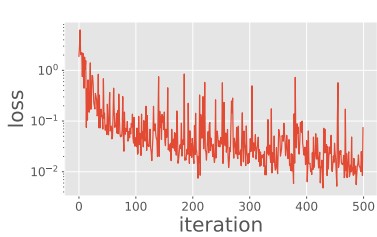

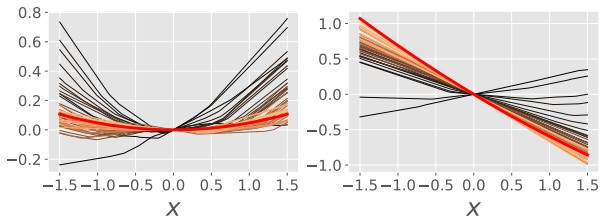

(a) Evolution of loss estimate over first 500 optimization iterations.

(b) Evolution of neural network $N_0(\mathbf{x}, \mathbf{y})$ (*black to copper*) approximating the initial value function $v(\mathbf{x}, \mathbf{y}, 0)$ (*red*) over first 500 optimization iterations for a typical (*left*) and an extreme (*right*) observation $y$.

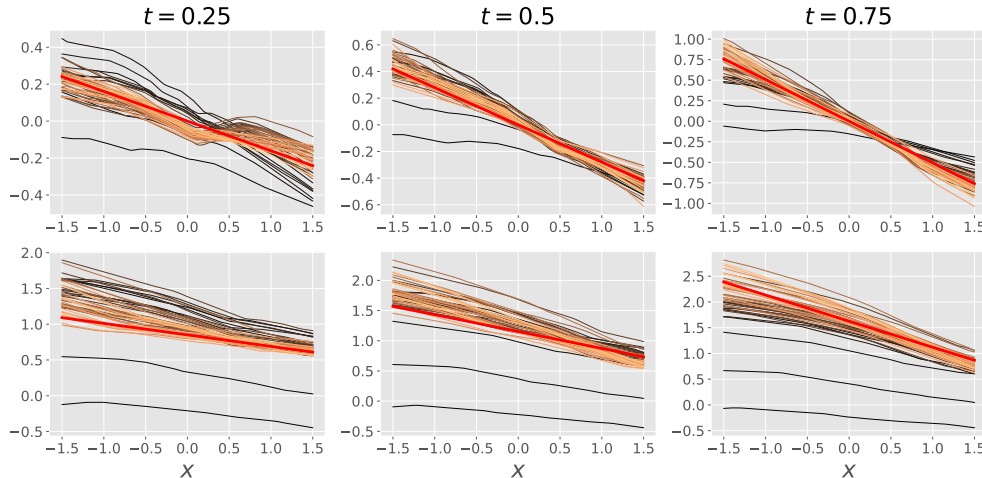

(c) Evolution of neural network $-N(\mathbf{x}, \mathbf{y}, t)$ (*black to copper*) approximating the optimal control function $\mathbf{c}_\star(\mathbf{x}, \mathbf{y}, t)$ (*red*) over first 500 optimization iterations for a typical (*upper row*) and an extreme (*lower row*) observation $y$.

Figure 5: Results for Ornstein–Uhlenbeck model with $d = 1$ and $\sigma_{\mathbf{Y}} = 1.0$ during initial training phase.

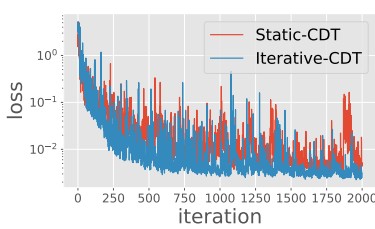

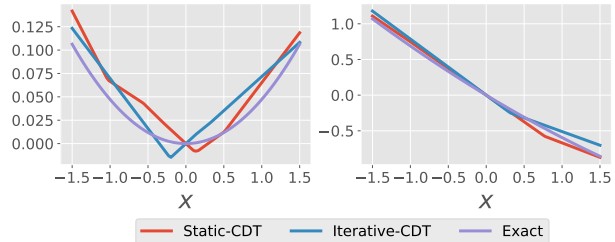

(a) Evolution of loss estimate over 2000 optimization iterations under static and iterative CDT schemes.

(b) Neural network approximation $N_0(\mathbf{x}, \mathbf{y})$ of the initial value function $v(\mathbf{x}, \mathbf{y}, 0)$ after training with the static and iterative CDT schemes for a typical (*left*) and an extreme (*right*) observation $y$.

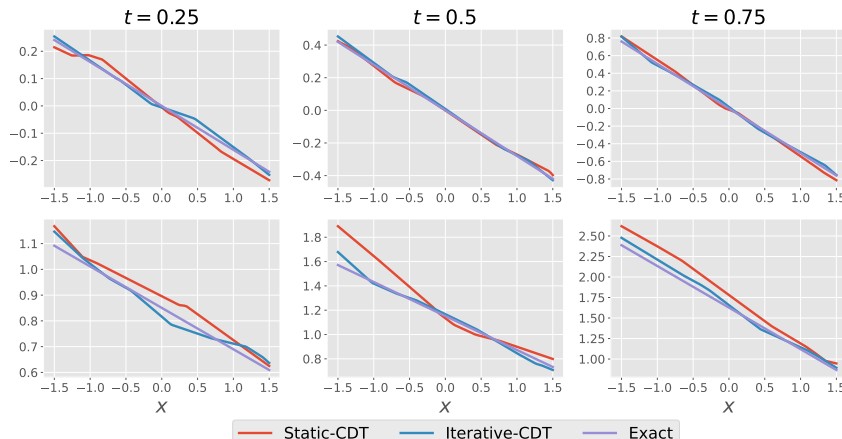

(c) Neural network approximation $-N(\mathbf{x}, \mathbf{y}, t)$ of the optimal control function $\mathbf{c}_\star(\mathbf{x}, \mathbf{y}, t)$ after training with the static and iterative CDT schemes for a typical (*upper row*) and an extreme (*lower row*) observation $y$.

Figure 6: Results for Ornstein–Uhlenbeck model with $d = 1$ and $\sigma_{\mathbf{Y}} = 1.0$ after training.

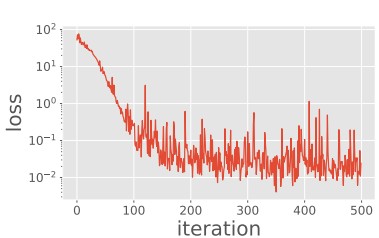

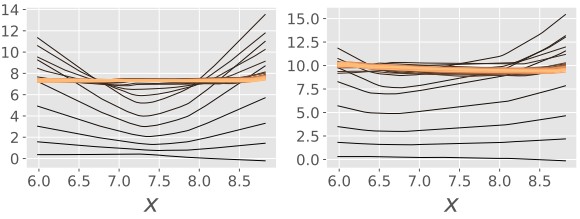

(a) Evolution of loss estimate over first 500 optimization iterations.

(b) Evolution of neural network $N_0(\mathbf{x}, \mathbf{y})$ (*black to copper*) approximating the initial value function $v(\mathbf{x}, \mathbf{y}, 0)$ over first 500 optimization iterations for a typical (*left*) and an extreme (*right*) observation $y$.

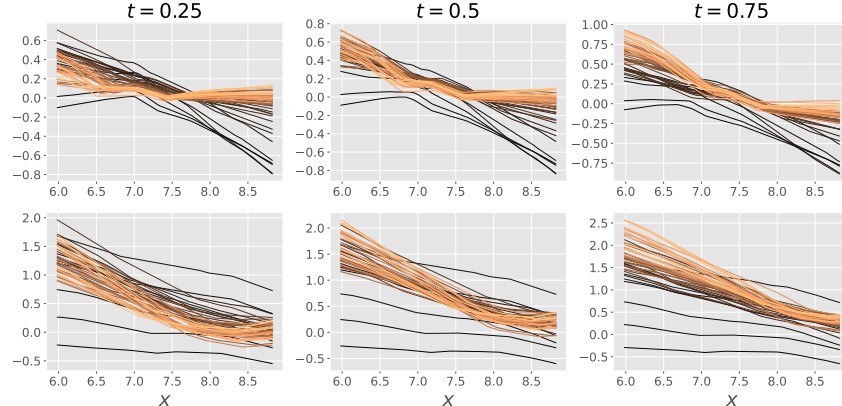

(c) Evolution of neural network $-N(\mathbf{x}, \mathbf{y}, t)$ (*black to copper*) approximating the optimal control function $\mathbf{c}_\star(\mathbf{x}, \mathbf{y}, t)$ over first 500 optimization iterations for a typical (*upper row*) and an extreme (*lower row*) observation $y$.

Figure 7: Results for logistic diffusion model with $\theta_4 = 1.069$ during initial training phase.

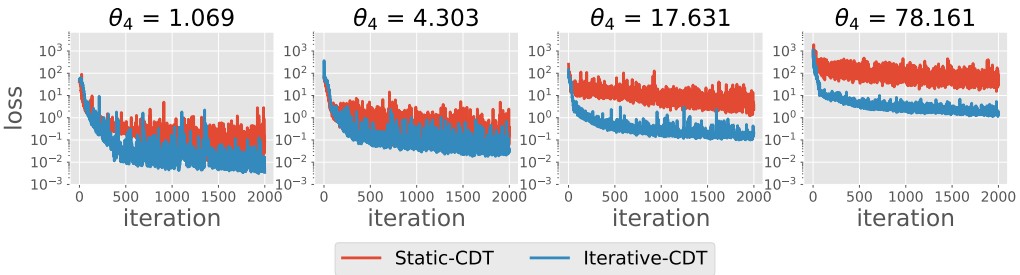

(a) Evolution of loss estimate over 2000 optimization iterations under static and iterative CDT schemes and various levels of informative observations.

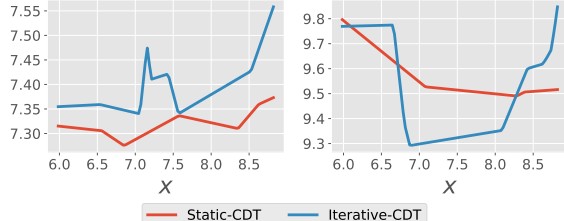

(b) Neural network approximation $N_0(\mathbf{x}, \mathbf{y})$ of the initial value function $v(\mathbf{x}, \mathbf{y}, 0)$ after training with the static and iterative CDT schemes for a typical (*left*) and an extreme (*right*) observation $y$.

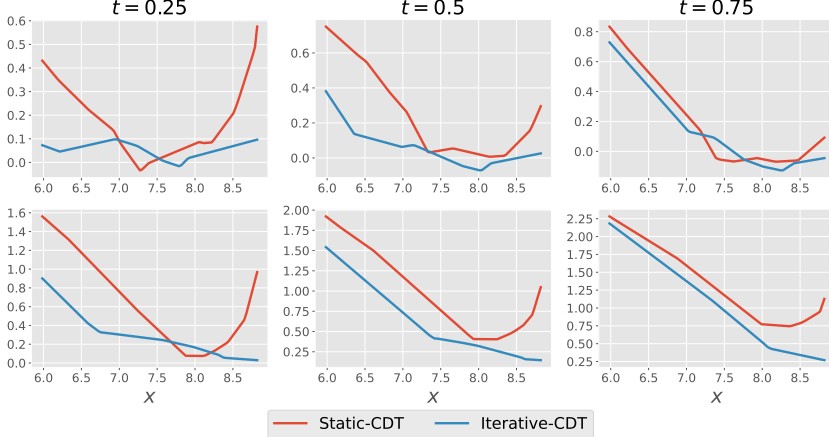

(c) Neural network approximation $-N(\mathbf{x}, \mathbf{y}, t)$ of the optimal control function $\mathbf{c}_\star(\mathbf{x}, \mathbf{y}, t)$ after training with the static and iterative CDT schemes for a typical (*upper row*) and an extreme (*lower row*) observation $y$.

Figure 8: Results for logistic diffusion model with $\theta_4 = 1.069$ after training.

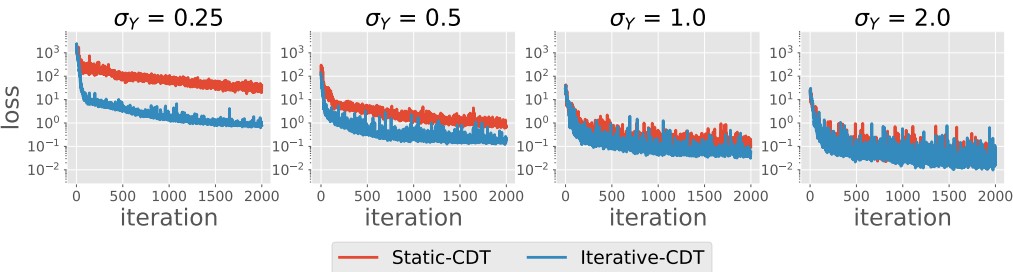

(a) Evolution of loss estimate over 2000 optimization iterations under static and iterative CDT schemes and various levels of informative observations.

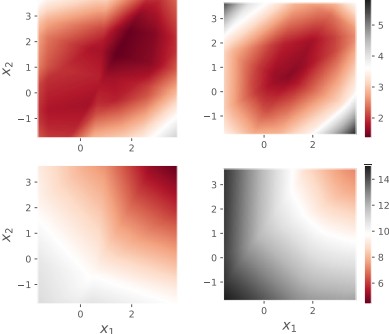

(b) Neural network approximation $N_0(\mathbf{x}, \mathbf{y})$ of the initial value function $v(\mathbf{x}, \mathbf{y}, 0)$ after training with the static (*left column*) and iterative (*right column*) CDT schemes for a typical (*upper row*) and an extreme (*lower row*) observation $y$.

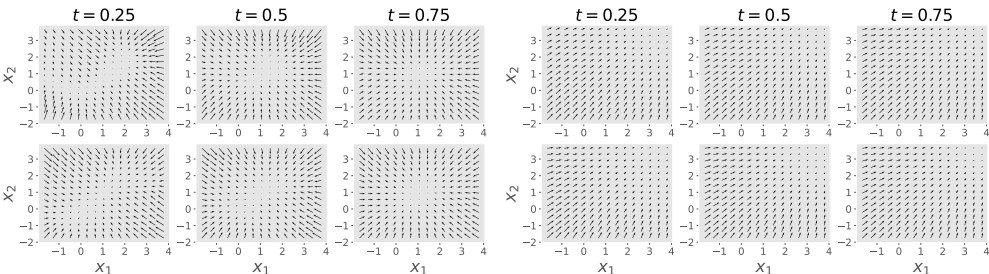

(c) Neural network approximation $-N(\mathbf{x}, \mathbf{y}, t)$ of the optimal control function $\mathbf{c}_\star(\mathbf{x}, \mathbf{y}, t)$ after training with the static (*upper row*) and iterative (*lower row*) CDT schemes for a typical and (*columns 1-3*) an extreme (*columns 4-6*) observation $y$.

Figure 9: Results for cell model with $\sigma_\mathbf{Y} = 0.5$ after training.

