# OpenReview forum: "Computational Doob h-transforms for Online Filtering of Discretely Observed Diffusions"
_ICLR.cc/2023/Conference — Submitted to ICLR 2023_

### Official Review · Reviewer_FHwj · 2022-10-25

**Confidence:** 3
**Correctness:** 3
**Technical Novelty And Significance:** 2
**Empirical Novelty And Significance:** 2
**Recommendation:** 5

**Clarity, Quality, Novelty And Reproducibility:**

The motivations and proposed methods are clearly expressed with rich equations. The experiments can be enriched to better align with real-world applications.
The reproducing ability is limited – to first fully understand the equations and then code on them is not a trivial work.


**Strength And Weaknesses:**

Strong

1. A novel computational framework with nonlinear feynman-kac formulas and neural networks to approximate these Doob’s h-transforms;

2. Detailed background, method and related work expressions to better learn the motivations and solutions of the major novel ideas;

Weak:

1. Currently, could find less information about the neural networks used and how representative learning is performed – this paper aligns more with statistics and applied mathematics;

2. More real-world datasets such as security pricing or portfolio changing experiments are preferred to be the application fields to better testify the ideas/algorithms proposed in this paper.


Detailed questions and comments:

1. Figure 1, 2, are curves – is it feasible to also list tables with specific numbers to learn the exact improvements (ranges) and differences?

2. The OU process (Ornstein-uhlenbeck) is frequently used in real-world domains such as finance, do we have further experiments on real-world datasets and comparisons with baseline algorithms?

3. ‘neural networks’ are mentioned around 30 times in this paper, prefer to learn the detailed architecture of the neural network and the complexity of it.

4. In the reference, I can hardly find machine learning related references or articles, or even representative learning related. So this paper is more suitable for statistics or mathematics.


**Summary Of The Paper:**

This paper deals with online filtering of discretely observed nonlinear diffusion processes which has rich applications in fields such as security pricing and modeling in finance. The authors propose a computational framework to approximate the Doob’s h-transforms that are typically intractable by solving the underlying backward Kolmogorov equations using nonlinear Feynman-kac formulas and neural networks.
Simulations and numerical experiments show that this approach can be orders of magnitude more efficient than sota particle filters in difficult regimes.


**Summary Of The Review:**

The overall solution to online filtering of discretely observed nonlinear diffusion processes is a promising direction in so many real-world applications. This paper provides approximations by leveraging neural network and Keynman-kac formulas which is interesting.

The work can be further improved by applying to real-world datasets and the network network’s architecture should be described and compared in detail for ablation study.

---

### Official Review · Reviewer_AGRQ · 2022-10-31

**Confidence:** 1
**Correctness:** 3
**Technical Novelty And Significance:** 3
**Empirical Novelty And Significance:** 3
**Recommendation:** 5

**Clarity, Quality, Novelty And Reproducibility:**

I had a hard time understanding what was going on in the paper, but I also lack background in this area.

**Strength And Weaknesses:**

Disclaimer -- I found this paper rather challenging to read -- it seems to be targeted towards a more applied math audience rather than a deep learning/ICLR audience and I am unsure of what exactly is being learned by the neural network / training objective.

# Strengths

**Significance.** This paper appears to solve a significant issue where we seek to effectively determine unobserved portions of a diffusion process.

**Clarity.** The paper seems to be relatively well written.

# Weaknesses

**Research Area.** I'm not sure if this paper is targeted to the ICLR audience. I had large amounts of difficulty reading the paper and could not understand what exactly was being learned.

**Clarity.** The paper has a large number of mathematical formula and was very dense to read. It may be nicer to provide a high level overview of what the paper is trying to as well as the existing challenges and what explicitly the work is trying to tackle.

**Summary Of The Paper:**

This paper studies how to simulate conditional diffusion processes -- where given some observations, we wish to study what the resultant state of the unobserved proportions of the diffusion process may be. To solve this issue, the authors propose a  Computational Doob's transform, leveraging neural networks to more accurately observed the proportions of the diffusion process

**Summary Of The Review:**

I don't really understand this paper -- I'm giving the paper a borderline accept but am complete unsure of my assessment.

---

### Official Review · Reviewer_rVdn · 2022-11-03

**Confidence:** 4
**Correctness:** 2
**Technical Novelty And Significance:** 2
**Empirical Novelty And Significance:** 2
**Recommendation:** 1

**Clarity, Quality, Novelty And Reproducibility:**

The paper is not written in a clear way, the reason to use the control formulation and going to the point of using neural networks to approximate it is not explained. In terms of novelty, this kind of work is quite commonplace in particle filtering. The authors shared their code anonymously, I think that is very good.

**Strength And Weaknesses:**

Strength: Getting good proposals within particle filtering is an important problem

Weaknesses: I think there were many (see below), mainly

(i) Lack of clear and coherent description of the problem
(ii) Experiments are low dimensional and are not interesting

**Summary Of The Paper:**

This paper proposes a new filtering methodology based on computational Doob's h-transform. The novelty is to get more efficient proposals, at least aim at it.

**Summary Of The Review:**

This paper has a number of problems before it can be published in a venue like ICLR. I think the current version is simply not interesting to the researchers in this field (ICLR crowd) - even to particle filtering people.

The paper phrases the problem of getting a good proposal as a control problem and approximates the relevant quantities using complex function approximators using neural networks. The claim is that the resulting method is more efficient under some extreme situations, like outliers or high-dimensions.

This paper could be an interesting contribution, it lacks very crucial experimental evidence:

(i) While there is a sentence about runtime being two minutes and being less costly than BPF with "many particles", this claim was not made quantitative by *showing* that this is the case. This is not how computational efficiency claims are made.

(ii) The dimensions are tested up to 32 are quite low. The claim in the earlier parts of the paper about efficiency in high-dimensions are completely unsupported. There are simply no true high-dimensional experiments, nothing like some of the published literature on PFs.

(iii) "Informative observations" were tested by changing likelihood variance but another claim in the abstract "when the observations are extreme under the model" was not tested properly. What kind of extremes are we talking about? Is there a certain statistical setting for these extremes? This was not in the main text and might be pushed somewhere in the appendix, despite it looks like there was lots of space to use in the main text (e.g. page 8).

Given that there is almost no convincing experimental result, I won't discuss theoretical setting. It looks like this paper was rushed for the deadline and there was no time for experiments. It definitely needs comprehensive and rigorous updating and is not suitable for publication at this stage.

---

### Official Review · Reviewer_dabH · 2022-11-03

**Confidence:** 4
**Correctness:** 3
**Technical Novelty And Significance:** 3
**Empirical Novelty And Significance:** 2
**Recommendation:** 5

**Clarity, Quality, Novelty And Reproducibility:**

Quality and Clarity. The paper is more-or-less well-written. Yet there are some questions and points to improve regarding the paper flow:

1) The paper's contributions should be explicitly stated somewhere at the beginning of the paper.

2) It seems that the equation (6) contains a misprint (as I understand). The conditional measure in the integral $p_{T-t}(d \boldsymbol{x}_T | \boldsymbol{x})$ should be substituted with $p_{t}(d \boldsymbol{x}_T | \boldsymbol{x})$

3) What are the distributions $\eta_{\boldsymbol{X}}$ and $\eta_{\boldsymbol{Y}}$ introduced on page 5?

4) It is clearer and more convenient to introduce the CDT algorithm as a listing with pseudocode.

5) The online filtering section also requires a listing with the proposed algorithm for nonlinear filtering.

6) I have some questions regarding metrics used to compare different nonlinear filtering models. How are they defined? In particular, it needs to be clarified how to estimate the marginal likelihood $\hat{p}(y_1, \dots, y_K)$? What is an Effective Sample Size? Another question: Why is $\mathbb{E}(\log \hat{p}(y_1, \dots, y_K)$ called ELBO?

Novelty: To the best of my knowledge, the proposed approach is definitely novel. The closest method is the bayesian bootstrap filter which is based on a much simpler mathematical foundation. The paper utilises some ideas from previous works, but their combination is rather unique and original.

Reproducibility: I didn’t run the code provided, but it seems to be clearly written, and I have no particular doubts regarding reproducibility.


**Strength And Weaknesses:**

Strength: The mathematical foundations of the proposed method are interesting, but the problem of nonlinear filtering also seems to have important applications.

Weaknesses: From my point of view, the paper under consideration has a relatively poor experimental section. Actually, the authors consider several synthetic nonlinear filtering problems and compute the metrics which only indirectly assess the quality of the proposed nonlinear filtering approach. Therefore, I still don't know if the method outperforms the competitive ones. I understand that it is difficult to find nonlinear filtering problems with known ground truth. That is why a better way is to consider some real-world nonlinear filtering applications with domain - specific metrics (probably not directly related to the nonlinear filtering itself) and evaluate the proposed approach on such a task.



**Summary Of The Paper:**

The paper proposes a mathematically-grounded method for solving nonlinear filtering problems in a particle-based manner. The method is based on Auxiliary Particle Filters and, compared to Bayesian Bootstrap Filter [1], requires substantial mathematical and ML engineering efforts to be practically implemented. The authors show the validity of the approach by modeling several synthetic nonlinear filtering routines.

[1] Salmond et. al. Novel approach to nonlinear/non-Gaussian Bayesian state estimation, 1993


**Summary Of The Review:**

The proposed method is theoretically interesting and mathematically rich. However, the applications section should be expanded with more convincing experiments, including real-world applications.

---

### Decision · Program_Chairs · 2023-01-20

**Decision:**

Reject

**Justification For Why Not Higher Score:**

As described in the weakness section, this paper has a problem in its experiments and writing. As long as these issues are not resolved, it is difficult to be published in ICLR. This paper requires major revision so that the proposal becomes more convincing.

**Justification For Why Not Lower Score:**

N/A

**Metareview: Summary, Strengths And Weaknesses:**

This paper gives a mathematically grounded new method to implement the Fully Adapted Auxiliary Particle Filter (FA-APF) that minimizes a local variance criterion. The method employs the so-called "Computational Doob’s h-Transform (CDT)" framework that relies on nonlinear Feynman-Kac formulas. To realize that, application of deep learning is proposed to approximate the optimal control function required to compute CDT. The validity of proposed method is justified by a simple numerical experiment.

Strength: The proposed method is new and it has a strong mathematical back-ground. Utilizing deep learning in the particle filtering method seems promising to apply the method to complicated and high-dimensional problems in several real world problems.

Weakness: On the other hand, the paper has weakness mainly in (1) the numerical experiment and (2) writing:
(1) The numerical experiment is quite weak. The problem setting of OU-process is considerably simple, and the state-space dimensionality is not high. It is expected to conduct experiments on more interesting practical (or difficult) problems (although they would be a bit difficult to obtain the ground truth). The computational efficiency of the proposed method is not fully investigated. The effectiveness of the proposed method to cope with an extreme case as stated in the abstract is not well justified.
(2) The writing can be much improved. Although the mathematical introduction is instructive, the algorithm can be introduced in a much more accessible way. Giving a pseudo-code would be more helpful. It is better if the authors more explicitly describe which function is approximated by neural networks and what kind of objective function should be optimized (the current writing somehow hides this point in a mathematically abstract description).